# An Observational Cohort Study on 194 Supraglottic Cancer Patients: Implications for Laser Surgery and Adjuvant Treatment

**DOI:** 10.3390/cancers13030568

**Published:** 2021-02-02

**Authors:** Gerhard Dyckhoff, Rolf Warta, Christel Herold-Mende, Elisabeth Rudolph, Peter K. Plinkert, Heribert Ramroth

**Affiliations:** 1Department of Otorhinolaryngology, Head and Neck Surgery, University of Heidelberg, 69120 Heidelberg, Germany; Christel.Herold-Mende@med.uni-heidelberg.de (C.H.-M.); Rolf.Warta@med.uni-heidelberg.de (R.W.); Peter.Plinkert@med.uni-heidelberg.de (P.K.P.); 2Heidelberg Institute of Global Health, University of Heidelberg, 69120 Heidelberg, Germany; Elisabethrudolph@web.de (E.R.); Heribert.Ramroth@uni-heidelberg.de (H.R.)

**Keywords:** laryngeal cancer, supraglottic, larynx preservation, transoral laser microsurgery, radiotherapy, chemoradiation, adjuvant therapy

## Abstract

**Simple Summary:**

Supraglottic laryngeal cancer patients suffer from a much poorer prognosis than patients with carcinoma arising from the glottis. Outstanding outcomes after laser surgical resection of early-stage supraglottic cancers have been reported by laser surgery centers of excellence. The aim of our retrospective observational cohort study was to assess whether similar results are achieved in the multi-institutional setting of less specialized facilities. We confirmed comparable oncological outcomes in five normal academic teaching hospitals, with a 5-year overall survival rate of 62.0% compared to 59.4–76.0% in the centers of excellence. In the case of microscopic residual tumors after surgical resection (R1) and/or lymph node metastases, adjuvant irradiation is recommended. Our data show that irradiation alone is not sufficiently effective in supraglottic cancers. Thus, for this tumor entity, adjuvant chemoradiation could be recommended.

**Abstract:**

Supraglottic laryngeal cancer is characterized by poor prognosis. In contrast, excellent outcomes have been published in early-stage supraglottic cancers after laser surgery in single-institutional series in centers of excellence. Are these results reproducible in the normal clinical practice of less specialized facilities? As part of an observational cohort study, the outcomes of 194 supraglottic cancer patients were assessed after treatment by larynx-preserving surgery (transoral laser microsurgery [TLM] or open partial laryngectomy [OPL]) or total laryngectomy (TL), with each having risk-adopted adjuvant treatment, or primary (chemo-)radiotherapy (pCRT or pRT). In early-stage supraglottic cancers, TLM achieved a 5-year overall survival (5-year OS) of 62.0%. No significant survival difference could be discerned between patients with and without adjuvant treatment (HR 1.47; 95% CI: 0.80 2.69). The comparison between pCRT and pRT patients suggests that CRT is more effective in supraglottic cancer. The 5-year OS rate achieved in our multiinstitutional setting is comparable to that reached in laser surgery centers of excellence (59.4–76.0%). According to our data and supported by the literature, adjuvant RT (aRT) is not sufficiently effective in supraglottic cancers. In case adjuvant therapy is indicated, adjuvant chemoradiation (aCRT) could be recommended.

## 1. Introduction

Laryngeal cancer is the most common head and neck carcinoma which, in turn, is the sixth most common cancer worldwide, with 177,400 new cases per year. The age-standardized (worldwide) incidence rates vary strongly between the sexes, with 3.6 and 0.5 new cases per 100,000 per year, respectively, and the age-standardized mortality rates are 1.9 and 0.3 per 100,000 in men and women, respectively [1]. Laryngeal cancers can be anatomically subdivided into glottic, supraglottic and subglottic cancers. A total of 60–70% originate from the glottis, and approximately 35% from the supraglottic site [2], supraglottic cancer occurs more frequently in women [3,4]. The main risk factors are smoking of tobacco and excessive intake of alcohol [5,6], with an even higher effect of alcohol in the supraglottic site [5,7]. As consistently shown, the outcome of supraglottic cancer is significantly poorer than that of glottic cancer. The 5-year overall survival (5-year OS) rate for localized disease is approximately 45–50% in supraglottic cancer compared to 85–90% in glottic cancer, indicating a 30–40% poorer 5-year OS in tumors of the supraglottis [4,8]. The poorer prognosis is thought to be a consequence of abundant lymphatic drainage of the supraglottic region, resulting in higher rates of regional and distant metastases [9,10]. Moreover, cancers of the glottis often cause hoarseness and thus are relatively often diagnosed at an early stage, whereas supraglottic and subglottic cancers more often present with nodal involvement at the time of diagnosis [11]. For the treatment of early-stage supraglottic cancers, the National Comprehensive Cancer Network (NCCN) guidelines recommend single-agent therapy with either larynx-preserving surgery alone (endoscopic resection or open partial laryngectomy) or definitive radiotherapy alone [12]. 

We report the oncological outcomes of the subgroup of supraglottic laryngeal cancers of a cohort of index laryngeal cancer patients of a defined study region in southwest Germany. Patients were treated using surgical or conservative treatment modalities according to the standards of the five independent centers of the region. 

## 2. Results

In total, 810 index laryngeal cancer patients were recruited for this cohort. Fifty-three patients were excluded because they either presented with synchronous second primary disease at first diagnosis (*n* = 12), received no treatment with curative intent (*n* = 28), or had an unknown tumor stage (*n* = 13). In the remaining 757 patients, there were 465 glottic (64%), 194 supraglottic (26%), 14 subglottic (2%), and 51 transglottic cancers (7%) and 33 patients with unknown localization of the primary tumor (4%). The percentage of women was higher in supraglottic (14%) than in glottic cancers (8%). The median age in supraglottic cancers was 60 years compared to 63 years in glottic cancers. The tumor stage at primary diagnosis was comparatively higher in supraglottic carcinoma (T1: 19%, T2: 35%, T3: 27%, and T4: 19%) than in glottic carcinoma (T1: 63%, T2: 25%, T3: 9%, and T4: 4%). Among supraglottic cancer patients, 104 (54%) were treated by larynx-preserving surgery (*n* = 96 [49.5%] TLM, and *n* = 7 [3.6%] OPL), *n* = 57 (29.4%) by TL, and *n* = 34 (17.5%) by primary conservative treatment (*n* = 20 [10.3%] pCRT, and *n* = 14 [7.2%] pRT). The median follow-up was 8.4 years (0.04–16.8 years) and included patients with recurrence and death. Table 1 gives a demographic and clinical overview of the supraglottic tumor patients within the different treatment arms. 

Early-stage (T1 and T2) supraglottic cancers after laser surgery showed a significantly worse OS than early glottic cancers (HR adjusted for age and comorbidities: 1.78, 95% CI:1.27–2.49; *p* = 0.0009) (Figure 1). The 5-year and 10-year disease specific survival (DSS) and overall survival (OS) Kaplan-Meier estimates and the corresponding 95% CIs of early-stage supraglottic and glottic cancers are shown in Table 2. 

Histopathologically clear resection margins (R0) were documented in 40.6%, tumor ramifications were microscopically described in 18.8% (R1), tumor was retained macroscopically (R2) in four patients (4.2%), definite R classification could not be established in 29.2% (Rx), and the resection status was missing in the charts of six patients (Table 3). A multivariable Cox regression model did not show any significant survival difference between uncertain margins (Rx) and histopathologically free margins (R0) (HR 1.41; 95% CI: 0.72–2.78) or between positive resection margins (R1, R2) and R0 (HR 0.71; 95% CI: 0.35–1.48). 

In the TLM group, aRT was administered in 43 (44.8%) patients, aCRT in 4 (4.2%), aCT in 2 (2.1%) and no adjuvant treatment in 47 (48.9%) patients. The distribution of the T category is given in Table 4. 

The procedural characteristics of neck dissection, resection status, and adjuvant treatment based on UICC stage are given in Table 5.

In a multivariable Cox regression model, no significant survival differences could be discerned between patients with and without adjuvant treatment (HR 1.47; 95% CI: 0.80–2.96). As an example of the evaluation matrix of radiation efficacy (definitions and set-up see Material and Methods) the results for T2 patients with clear indication for adjuvant therapy is shown in Table 6. 

For the sake of maximum power without restriction to TLM, a multivariable Cox regression model of the whole cohort of 194 supraglottic cancer patients was performed comparing the outcome of all surgically treated patients (TLM, OPL, and TL) with that of conservative patients. There was a significantly worse outcome after pRT alone (HR 2.39; 95% CI: 1.33–4.29) but there was no significant difference between pCRT and surgery (HR 1.53; 95% CI: 0.93 2.52) (Figure 2).

## 3. Discussion

As consistently shown, supraglottic cancers have a poorer prognosis than glottic cancers [4,8,13]. As an example, Silvestri reported a 3-year adjusted survival rate of 46% for supraglottic carcinoma patients compared to 83% for glottic carcinoma patients [13]. One reason is that glottic tumors are diagnosed at an earlier stage than tumors of the supraglottic site, as hoarseness, the cardinal symptom of glottic carcinoma, occurs with few lesions of the vocal folds. In our cohort, 62.4% of glottic tumors were diagnosed as early as stage I, compared to no more than 18.6% in supraglottic cancers. This is consistent with Raitiola, who found 51.2% stage I glottic tumors compared to 14.2% T1 tumors arising from the supraglottis [11]. A second important reason is that supraglottic cancers more often present with nodal involvement at the time of diagnosis [8,11,13]. This is ascribed to the richer lymphatic supply of the supraglottic area [9,10]. Thus, even in T1 and T2 supraglottic carcinomas, treatment of the neck has to be considered. In our cohort, 55.3% of T1 patients and 70.0% of T2 patients received neck dissection. A multivariable Cox regression model also revealed a poorer prognosis for supraglottic cancers in our cohort. Focusing on early-stage, we found a significantly worse OS after TLM in supraglottic cancers than in glottic carcinomas (HR 1.78, 95% CI:1.27–2.49; *p* = 0.0009) (Figure 1). Iro et al. published a comparable difference in outcome (5-year OS of 59.4% in supraglottic tumor patients compared to 77.7% in glottic carcinoma patients) [14]. 

After important pioneering work was performed by Vaughan and Davis using a carbon dioxide (CO_2_) laser for resection of benign and malignant lesions of the supraglottis first [15,16], Steiner systematically introduced TLM for the curative treatment of supraglottic cancers in Germany [17]. As an excellent surgeon and teacher, he established the laser surgery schools of Goettingen and Erlangen-Nuremberg, which performed several single institution series with outstanding results [14,17,18,19,20]. A similar outcome was achieved by Davis in Salt Lake City (UT, USA) [21]. The question is whether these results are representative of TLM in the multiinstitutional setting of less specialized centers. As an answer, we report the outcomes of laser surgery in supraglottic tumors as part of a retrospective observational cohort study of all index laryngeal cancer patients of a defined study region who were treated using surgical or conservative treatment modalities according to the standards of five independent academic teaching hospitals. The 5-year and 10-year OS rates were 65% and 32% in T1 and 61% and 45% in T2 supraglottic cancer patients, respectively. In most published studies in the literature, OS is presented for early stage (T1 and T2) patients together and mostly as 5-year OS. The 5-year OS of our pooled T1 and T2 patients (*n* = 70) amounted to 62.0%. In Table 7, the 5-year OS of our observational multicenter study fits the results achieved by the centers of excellence in which the OS rate ranges from 59.4% to 76%. Notably, the OS reported for combined stage I + II disease is probably better than the OS reported for pT1 + pT2 patients. This is because in the stage I + II group, all pT1 and pT2 tumor patients, who suffered from cervical lymph node metastases and thus were classified stage III or IV, were excluded.

A direct comparison of some further important work is not possible because only 3-year, but not 5-year OS was given (e.g., Rudert et al.: 17 patients T1 + T2: 3-year OS: 88% [23]), or higher stage tumors were included (e.g., Davis: 46 cT2 patients, 39% histopathologically restaged from cT2 to pT3 because of preepiglottic space infiltration, 5-year OS: 63% [21]). This work, however, fits in the range of outcomes published by the other centers of excellence. From the epidemiological point of view, a direct comparison of these studies was prohibited, as the Kaplan-Meier estimates are univariate and do not consider differences in the composition of the respective cohorts as to age, comorbidities, or other confounders. Compared with the other publications, our study is the only one on supraglottic cancers providing long-term results after laser surgery with 10-year OS data.

While the oncological outcomes are comparable, great discrepancies can be seen between the results of our five regional facilities taking part in our observational cohort study and the results of the single-institution series published by centers of excellence when considering the resection status. In 2018, Ambrosch reported outstanding 90.1% successful R0 resections with one single positive microscopic tumor ramification. In only 8.8% (*n* = 8), definite R classification could not be established [20]. In 2011, Iro et al. reached the same excellent result of 91.2% R0 resections and only 8.8% R+. The number of R+ patients was too low to enable statistical interpretation in the form of log-rank *p*-values [14]. These outstanding results were reached in centers of excellence by highly experienced and very well-trained laser surgeons. Iro emphasizes the importance of certain R0 resection to achieve satisfactory oncological results [18]. Optimal exposure and visualization of the tumor is necessary for safe tumor resection, as Steiner and Ambrosch point out in their description of the operative technique [17]. Of exceeding importance for proper histopathological assessment of the resection margins is the careful orientation and accurate mapping of the resected specimens, especially if the resection has to be performed in several blocks (usually necessary in advanced tumor stage) [24]. Histopathological examination must be performed by serial sections. For the achievement of accurate pT staging, it must finally be possible for the surgeon “to recreate a complete 3-dimensional mosaic of the entire lesion” [24]. If no clear R0 resection is reached, reresection is necessary. If this is not possible the switch to transcervical open surgery or even TL is indicated [17,18,21]. 

In contrast to these outstanding results of up to over 90% R0 resection and Rx of less than 10% in the centers of excellence, the evaluation of our multicenter study revealed histopathologically proven R0 resections in only half of the cases (T1: 45%, and T2 51%), while there was a large number of Rx (T1: 32%, and T2: 21%) and R1 resections (T1: 16%, and T2: 23%). However, multivariable Cox regression analysis revealed no significant difference in outcome between uncertain margins (Rx) and histopathologically free margins (R0) (HR 1.41; 95% CI: 0.72–2.78), or between positive margins (R1, R2) and R0 (HR 0.71; 95% CI: 0.35–1.48). This is in accordance with a recent database study of Jumaily et al., who analyzed 747 T1–2 glottic carcinomas treated by TLM. 5-year OS for patients with positive margins was lower (80.0%) than that with negative margins (82.9%), but without statistical significance in multivariate analysis (*p* = 0.96). The rate of positive resection margins in his study was 19.9% [25]. Regarding the more favorable survival data, it should be noted that the study was conducted on glottic carcinomas and 89.1% of patients presented with T1 tumors. In a second recent database study on 1959 TIS-T3 laryngeal cancer patients (of which 21.3% were supraglottic), Hanna reported 31.3% positive margins in supraglottic cancers compared to 19.7% in glottic tumors, without showing statistical significance in multivariable analysis [26]. However, no survival data were presented. In a series of 203 laryngeal cancer patients with 39% supraglottic tumors, Pedregal-Mallo reported 17% positive margins in supraglottic tumors compared to 2% in glottic cancers [27]. In his study, involvement of surgical margins was associated with lower disease-specific survival in multivariable analysis (*p* = 0.004) [27]. Interestingly, in the large database studies of Hanna and Jumaily no Rx or uncertain margins were reported [25,26]. Ambrosch and Steiner, however, point out that this is a realistic situation to face even when following the highest standards. If tumor infiltration reaches cartilage or bony structures, clear R0 resection may be impossible to prove; in these cases, an Rx status cannot be omitted [24]. Thus, in most European studies beside R1, R2, and R0 a certain amount of Rx is also reported. The question is whether “Rx/uncertain resection margin“ was actively captured in the National Cancer database. If an information specialist entering data can only choose between R0 and R1 but a resection margin is uncertain, documentation bias can occur. Therefore, the information given by Hanna is extremely interesting: only 55% of patients with positive margins received adjuvant radiation [26]. Follow-up of the other 45% could prove the correct assessment of positive margins (R1). In true residual tumors, recurrence and tumor related death are likely to occur. While Rx seems to be missing in these American studies, the Italian group of Fiz et al. used a sophisticated R classification system with 6 categories to differentiate between negative, close superficial (i.e., <1 mm), close deep, positive single superficial, positive multiple superficial, and positive deep [28]. Using this system, only 232 out of 507 glottic cancer patients were negative (45.8%), while the rest were close margin or positive. In this differentiated system, an undifferentiated Rx status appears but superfluous. The authors show that there is a significant correlation between the different forms of positive margins and recurrence-free survival. Interestingly, this study did not follow Steiner’s dogma that the paramount aim of laser surgery is achieving histopathologically free margins. Patients with single positive superficial margins—the same as with close margins—were strictly followed-up but not reresected. Only in positive multiple superficial or deep margins was further treatment (TLM reresection, OPL, or aRT) proposed to the patient. Using the recurrence-free survival (RFS) of 89.4% of the pooled cases of negative and close margin cases as a reference, single superficial had an RFS of 83.3% (*p* = NS), multiple superficial 72.7% (*p* < 0.001) and deep positive margins 75.8% (*p* < 0.01) [28]. 

In 1998, with the beginning of systematic TLM in supraglottic cancer, Iro published the first series with an R0 resection rate in T1 and T2 supraglottic tumors of 88.4% (61/69 patients) in a study of 141 patients with an overall R0 resection rate of 78% [18]. With the experience of the corresponding R+ patients, Iro reported a clear correlation between resection status and survival. With a value of *p* < 0.001, the difference in survival between patients with negative margins (R0) and those with positive margins despite receiving aRT, reached a high significance level (OS of 76.4% compared to 17.8%) [18]. The question remains why there was no statistical correlation between poor resection status and survival—neither OS nor DSS—in our multiinstitutional study. Furthermore, the patients in our study with an R0 resection rate of 48% had a survival outcome of 62%, which is comparable to the 59.4% in the study of Iro, in which an R0 resection was achieved in more than 90% of cases. Meanwile, in Iro’s study, there was a statistically significant correlation between resection status and outcome. The solution might be found in the Rx resections. According to Ambrosch and Steiner, Rx status is properly assessed in rare cases in which deep tumor infiltration reaches cartilage or bony structures. However, in all other cases, after properly applying Steiner´s technique of careful orientation and accurate mapping of the resected specimen “to recreate a complete 3-dimensional mosaic of the entire lesion,” a distinct R0 or R1 resection should be possible [24]. If fresh frozen sections reveal residual tumor, the resection has to continue until clear margins are reached and the pathologist reports no more ramification in the last properly oriented specimen. It could be assumed that this meticulous work of careful orientation, accurate mapping and recreating the mosaic is not done in the same scrupulous way in less specialized facilities as in a center of excellence. Therefore, resection may be performed in an excellent manner, reaching clear margins with consequent favorable survival outcomes—but the histopathological results do not reflect the surgical results, as the definite R classification cannot be established, and is thus called “Rx”.

On multivariable analysis, Hanna reported factors associated with positive margins. For T-stage T3 he found the highest negative impact with an HR of 5.53 (3.55–8.63) [26]. In parallel, in the T3 tumors of our cohort we reached R0 resection in only half of the number of cases compared to T1 and T2. In contrast, Hanna also found a smaller but still significant difference for T2 compared to T1 (HR 2.74 [2.05 3.65]), while the rate of R0 resection in our cohort was almost the same (T1: 45%, T2: 51%). A possible explanation might be found in the other negative impact factors described by Hanna: nonacademic vs. academic status, and lower caseload facilities had a higher likelihood of positive margins. Overall, 60.6% of patients reported in the National Database study were treated at facilities performing < 2 cases per year [26]. In contrast, all the centers in charge of oncological therapy in our study region were academic teaching hospitals performing laser surgery on a regular basis. Thus, our results can be seen in accordance with Hanna´s findings. Similarly, Ambrosch and Iro recommend TLM in supraglottic tumors as a good option for surgical larynx preservation for early supraglottic carcinomas (T1 and T2) but restrict this recommendation in locally advanced cancers—conforming to the current NCCN guideline [12]—to “only selected cases” [14,20]. They state this as representatives of laser surgery centers of excellence. Our observational multicenter study shows that ordinary academic teaching hospitals can achieve comparable results. Thus, we agree with Hanna that for T3, the proper patient selection for TLM is important, but for T1 and T2, it is more important to select the appropriate facility. The consequence of Hanna´s study should not be a change in the therapy concept but rather an improved training of laser surgeons and concentration of cases of supraglottic cancer in centers with sufficient experience and caseload, as proposed by Harréus [29]. Based on the experience of our study we would like more emphasis to be placed on the training and experience in laser surgery as well as on the accuracy in the processing of resected specimens because proper assessment of resection margins is required for the correct indication of reresection or adjuvant treatment.

In early-stage laryngeal cancer, the treatment of choice is single agent therapy, either surgical or nonsurgical. The paramount aim of the surgical approach is the complete removal of the tumor so that adjuvant therapy in node-negative disease is not necessary. Just as salvage surgery is an integral part of the conservative concept but primarily by no means intended, adjuvant treatment is a salvage option when the primary treatment goal is not reached, but further resection is not possible or not accepted by the patient. Thus, an adjuvant treatment rate of 51% in a TLM group must raise concern. The procedural characteristics based on UICC stage show, however, that only one stage I patient (4,5%) and nine stage II patients (23.7%) received adjuvant treatment because of a lack of free margins (Table 5). Most of the patients received adjuvant treatment because locoregional advanced stages.

The distribution of primary treatment modalities clearly reveals that, in the respective region of Germany 20 years ago, there was a strong preference for primary surgery over p(C)RT. This can be a source of bias, as patients with comorbidities, lower compliance or lack of social support might have been chosen for nonsurgical treatment. While the documented Charlson Comorbidity Index does not show this tendency, this possible bias could not be excluded in this retrospective study. Thus, one should exercise caution when comparing surgical and non-surgical treatment approaches. Moreover, the low number of patients treated conservatively compared to the large number treated by surgery makes a proper statistical comparison difficult. Refraining from inferential statistics concerning the outcome of TLM and pRT, valuable—even though preliminary—conclusions can be drawn concerning the effectiveness of radiotherapy alone compared to radiation with adjuvant chemotherapy. A Cox regression model comparing each—pCRT and pRT—to surgery showed an adjusted hazard ratio of 1.53 (95% CI: 0.93–2.52) which is not significant for pCRT compared to a statistically significant HR of 2.39 (95% CI: 1.33–4.29) for pRT. Since a direct comparison of pCRT and pRT is not reasonable because of the low number of cases, this allows, however, a cautious conclusion to be drawn concerning the superiority of CRT over RT in the entity of supraglottic cancer. Comparison of the Kaplan-Meier curves of the patients treated by pCRT and by pRT, descriptively illustrates the superiority of CRT over RT alone (Figure 2).

A multivariable Cox regression model revealed no significant survival difference between patients with and without adjuvant treatment (HR 1.46; 95% CI: 0.80–2.69). In an attempt to understand what “missing effect of adjuvant treatment” means on a differentiated and personalized basis, we designed an evaluation matrix of adjuvant treatment. At first, we defined criteria for the assessment of success and failure of adjuvant treatment (see Materials and Methods, Table 8). Patients were then classified according to the safety degree of indication of adjuvant treatment (no indication, uncertain, probable, and clear indication according to R status and N stage following the NCCN guidelines [12] and categorized of one particular T stage in a bidimensional matrix according to safety of indication and success of adjuvant treatment in terms of oncological outcomes given by survival [years, months] and cause of death. As an example, we present the results of the T2 patients with clear indication of adjuvant treatment (R1, R2, and/or N > 1) in Table 6. A direct comparison of the aRT patients with oncological success and oncological failure shows at first sight, that for the very same oncological indication the failure-patients outweigh the success-patients (*n* = 9 vs. *n* = 7). Considering the individual time to death by tumor (“tu”) gives an impact of the necessity to find a more effective adjuvant treatment to possibly prevent this outcome. The result of our study is consistent with Iro, who reported that adjuvant irradiation did not show a significant effect on the prognosis of patients with R+ resections (*p* < 0.001). The recurrence-free survival rate of patients in whom TLM did not induce an R0 condition was 17.8%, compared to 76.4% in R0 patients despite aRT [18]. Thus, as aRT could not compensate for positive resection margins, Iro´s consequence was to aim at R0 resection at all costs: either by (repeated) TLM reresection or by switch to OPL or—if necessary—by TL. If the patient did not agree to further surgery, there seemed to be no alternative. The data of our study, however, propose a different option: if adjuvant radiotherapy alone is ineffective in supraglottic cancer, efficacy can be augmented by the addition of chemotherapy. 

For the definitive setting, chemoradiation is an established treatment concept for advanced stage disease. For the adjuvant setting, it has become standard since the large prospective studies of Cooper and Bernier [30,31]. With level I evidence, they have shown improved survival in high-risk patients with positive resection margins (R+) and/or extracapsular spread (ECS+) in lymph node metastases for head and neck cancers (HNSCC) in general. In Cooper´s and Bernier´s cohorts, most tumors were oropharyngeal carcinoma (in the combined therapy group 48% and 32%, respectively), for which a high response rate for radiotherapy and especially chemoradiotherapy is well-documented. According to our data and supported by the literature, the entity of supraglottic cancer is not sufficiently responsive to adjuvant radiotherapy. In Cooper´s study, there were 15% and 14% supraglottic cancers in each treatment arm [30], and 23% and 22% laryngeal cancers in Bernier´s cohort, respectively, without explicitly mentioning the proportion of supraglottic cancers [31]. However, neither study showed specific survival data for supraglottic cancer. Evaluating HNSCC from different primary sites and with a preponderance of tumors with a high response rate, the irresponsiveness of an underrepresented entity could have been blurred. To our knowledge, our study shows for the first time for the specific entity of supraglottic cancers—although with the given precaution because of the rather low number of cases—the superiority of CRT over RT.

In several reports and reviews such as that of Silver et al., TLM with or without aRT is proposed as the preferred treatment for early supraglottic cancer [32]. Several studies have compared TLM and primary RT. To our knowledge, however, no study to date has proven the efficacy of adjuvant RT in supraglottic cancer patients by comparing TLM with and without aRT. There is, however, evidence from Iro et al. and from our present study, that aRT is not sufficiently efficacious in supraglottic cancer patients. While Cooper and Bernier have shown the superiority of aCRT over aRT for HNSCC in general, our data show—although with a rather small number of patients—that aCRT might be a treatment option superior to aRT for the specific entity of supraglottic cancer patients with positive margins who refuse further surgery and for patients with positive nodal disease (N+). In the NCCN guidelines, the option to “consider” systemic therapy/RT for supraglottic cancer patients with positive margins is given, while aRT is regarded as the therapy of choice. aCRT with concurrent cisplatin, however, is recommended for locally advanced disease [12]. According to Cooper and Bernier, for HNSCC in general, there is a survival benefit for high risk patients, leading to the recognized recommendation of aCRT in patients having R+ or ECS+. In our data, less than half of the failure patients had a positive resection status (R1) and in none of the patients positive extracapsular spread was reported. Thus, in the entity of supraglottic cancers, patients could experience a survival benefit even if these standard high-risk criteria are not given. Therefore, provided that the patient is fit enough, we suggest chemoradiotherapy for supraglottic cancers in the primary setting at the early stage and in the adjuvant setting for cases at high risk both in the sense of R+ or ECS+ and simply for belonging to this tumor entity with poor prognosis. However, the presumed survival benefit has to be verified in prospective clinical studies.

In accordance with the current guidelines, Ambrosch states that a single-modality therapy with the goal of curing the disease while preserving laryngeal functions is recommended. By applying quality of life (QoL) questionnaires, she convincingly proved that long-term swallow function and voice-related QoL are worse after multimodality treatment [20]. The addition of systemic chemotherapy will further increase the toxicity of adjuvant irradiation. Therefore, our recommendation should be validated in prospective studies with larger numbers of patients.

## 4. Materials and Methods

As reported previously [33], we recruited all index laryngeal cancer patients from the five centers in a well-defined region in southwest Germany that are in charge of the treatment of head and neck cancer patients between 5 January 1998, and 31 December 2004, resulting in a cohort of 810 patients. Patients for this study were identified in two different ways. The first part of the patient cohort was patients who took part in a previous prospective case-control study between 1998 and 2000. For the second part (2001–2004), all patients were identified retrospectively. The charts of all patients were evaluated retrospectively. A follow-up was performed up to March 2015 leading to a total follow-up time of 11 to 17 years. Demographic data and clinical information were obtained from hospital medical records using a standardized form. Vital status as well as date and cause of death were inquired from local registries. Retrospectively, the different therapy groups were checked for comparability with the established confounders sex, age, comorbidities, TNM stage, differentiation, and primary tumor site. OS rates were calculated using the Kaplan-Meier method. Regression analysis was performed using multivariable proportional hazards models. The OS rates of pCRT and pRT, both with the option of salvage total laryngectomy, were compared with those of surgery with adjuvant radiotherapy or adjuvant chemoradiotherapy, as indicated by risk and stage (OP ± a[C]RT). Survival time was defined as the time from the first diagnosis until death or until 21 March 2015. For the evaluation, patients who moved away from Germany were censored after one month of emigration. *p*-values below 0.05 were regarded as statistically significant. Only overall and disease specific survival estimates are shown. The following variables showed an effect in the univariate analysis (*p* < 0.20) and were included in the multivariable analysis as explanatory variables: age at first diagnosis (continuous), comorbidities, tumor site, TNM status, and therapy modality. Backward selection was used to receive a final model. The proportional hazards assumption was checked by adding a time-dependent version of all the variables in the model [34]. The assumption was met for all variables. The M status could not be evaluated as evidence of distant metastases could be proven for only 5 patients. Comorbidity conditions were defined using the Charlson Comorbidity Index (CCI), which summarizes 18 different comorbidities, weighted by severity, in a single score [35]. For this analysis, we used the binary form of the variable, which is set to one for CCI values of two or higher. The data analysis for this paper was generated using SAS/STAT software, Version 14.2 of the SAS System for Windows, copyright © 2012 (SAS Institute Inc., Cary, NC, USA). For differentiated evaluation of the efficacy of aRT, criteria for the assessment of success and failure of adjuvant treatment were defined. (See Table 8) Patients were classified as per safety degree of aRT indication (no indication, uncertain, probable, and clear indication) according to R status and N stage and categorized as per oncologic outcome in an evaluation matrix. 

## 5. Conclusions

In the multi-institutional setting of our observational cohort study, oncological outcome was achieved, which is comparable to that reached in laser centers of excellence. Laser surgery should, however, be performed in academic teaching hospitals or equivalent facilities with sufficient experience and caseload. Adequate care should be exercised in the processing of resected specimens to avoid Rx results. The paramount aim is the clear R0 resection. If free margins are not achieved by TLM, the switch to OPL is recommended. 

According to our data and supported by the literature, adjuvant radiotherapy alone is not sufficiently efficacious. There is, however, evidence that chemoradiation is more effective even in supraglottic cancer. Thus, we suggest chemoradiotherapy for supraglottic cancers in the primary setting even at the early stage and in the adjuvant setting for cases at high risk both in the sense of R+ or ECS+ and simply for belonging to this tumor entity with poor prognosis—provided that the patient is fit enough and wishes for maximum safety even at the cost of higher toxicity. However, further prospective studies with larger numbers of patients are warranted.

## Figures and Tables

**Figure 1 cancers-13-00568-f001:**
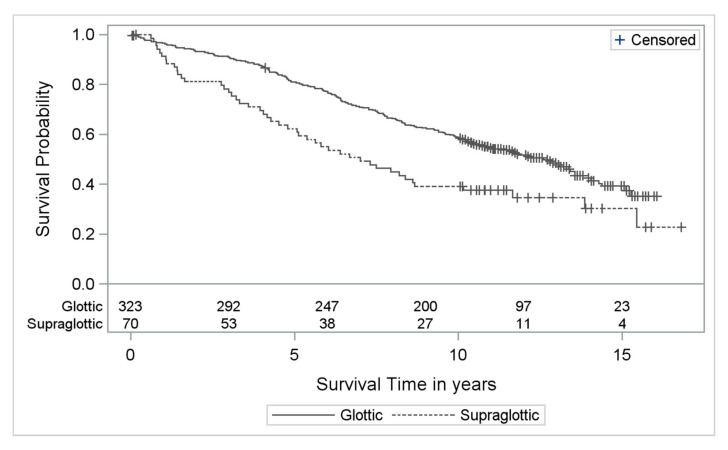
Kaplan Meier curves with numbers at risk of supraglottic compared to glottic cancers (T1 + T2) treated by TLM (OS) (solid line = glottic; dashed line = supraglottic).

**Figure 2 cancers-13-00568-f002:**
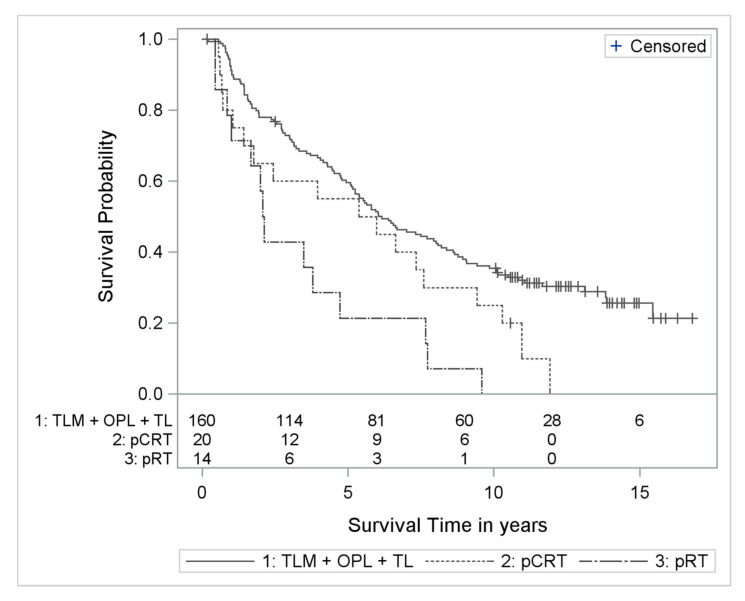
Kaplan Meier curves with numbers at risk of supraglottic cancers treated with pCRT or pRT compared to TLM + OPL + TL (OS).

**Table 1 cancers-13-00568-t001:** Demographic and clinical characteristics of 194 supraglottic cancer patients.

Variable	Category	TLM	OPL	TL	pCRT	pRT	All
Total		96	7	57	20	14	194
Age (cont.) ^a^		59.6 (37–79)	61.3 (51–77)	59.7 (40–80)	58.4 (41–77)	62.1 (40–79)	59.7 (37–80)
Sex	Males	81 (84.4)	7 (100)	53 (93.0)	15 (75.0)	10 (71.4)	166 (85.6)
	Females	15 (15.6)	0 (0.0)	4 (7.0)	5 (25.0)	4 (28.6)	28 (14.4)
CCI	0	71 (74.0)	4 (57.1)	38 (66.7)	16 (80.0)	5 (35.7)	134 (69.1)
	1	25 (26.0)	3 (42.9)	19 (33.3)	4 (20.0)	9 (64.3)	60 (30.9)
T category	1	31 (32.3)	1 (14.3)	1 (1.8)	2 (10.0)	2 (14.3)	37 (19.1)
	2	39 (40.6)	4 (57.1)	14 (24.6)	5 (25.0)	6 (42.9)	68 (35.1)
	3	16 (16.7)	2 (28.6)	24 (42.1)	7 (35.0)	3 (21.4)	52 (26.8)
	4	10 (10.4)	0 (0.0)	18 (31.6)	6 (30.0)	3 (21.4)	37 (19.1)
N stage	0	50 (52.1)	5 (71.4)	30 (52.6)	9 (45.0)	4 (28.6)	98 (50.5)
	1	15 (15.6)	0 (0.0)	7 (12.3)	2 (10.0)	3 (21.4)	27 (13.9)
	2	26 (27.1)	2 (28.6)	20 (35.1)	6 (30.0)	6 (42.9)	60 (30.9)
	3	1 (1.0)	0 (0.0)	0 (0.0)	2 (10.0)	1 (7.1)	4 (2.1)
	X	4 (4.2)	0 (0.0)	0 (0.0)	1 (5.0)	0 (0.0)	5 (2.6)
UICC stage	I	20 (20.8)	0 (0.0)	1 (1.8)	1 (5.0)	0 (0.0)	22 (11.3)
	II	20 (20.8)	4 (57.1)	7 (12.3)	3 (15.0)	4 (28.6)	38 (19.6)
	III	23 (24.0)	1 (14.3)	19 (33.3)	6 (30.0)	3 (21.4)	52 (26.8)
	IV	33 (34.4)	2 (28.6)	30 (52.6)	10 (50.0)	7 (50.0)	82 (42.3)
Neck diss	No	25 (26.0)	1 (14.3)	3 (5.3)	18 (90.0)	14 (100)	61 (31.4)
	Yes	71 (74.0)	6 (85.7)	54 (94.7)	2 (10.0)	0 (0.0)	133 (68.6)
Resection	0	39 (40.6)	6 (85.7)	38 (66.7)	0 (0.0)	0 (0.0)	83 (42.8)
	1	18 (18.8)	0 (0.0)	7 (12.3)	0 (0.0)	0 (0.0)	25 (12.9)
	2	4 (4.2)	0 (0.0)	0 (0.0)	0 (0.0)	0 (0.0)	4 (2.1)
	X	28 (29.2)	1 (14.3)	9 (15.8)	2 (10.0)	3 (21.4)	43 (22.2)
	missing	7 (7.3)	0 (0.0)	3 (5.3)	18 (90.0)	11 (78.6)	39 (20.1)
Adj. ther.	None	49 (51,0)	4 (57.1)	26 (46)	n.a.	n.a.	79 (40.1)
	aRT	43 (44.8)	3 (42.8)	23 (41.1)	n.a.	n.a.	69 (35.6)
	aCRT	4 (4.1)	0 (0.0)	7 (12.5)	n.a.	n.a.	11 (5.7)
	aCT	2 (2.0)	0 (0.0)	0 (0.0	n.a.	n.a.	2 (1.0)
	unknown	14 (14.6)	0 (0.0)	5 (8.8)	2 (10.0)	2 (14.3)	23 (11.9)

^a^ Mean (Standard Deviation); Age (cont.) = Age (continuous); Neck diss = Neck dissection; Resection = Resection status; Adj. ther. = Adjuvant therapy; n.a. = not applicable.

**Table 2 cancers-13-00568-t002:** 5-year and 10-year DSS and OS of early-stage supraglottic and glottic cancers after TLM.

Localization, T (*n*, DSS/OS) *	5-y DSS [%] (95% CI)	10-y DSS [%] (95% CI)	5-y OS [%] (95% CI)	10-y OS [%] (95% CI)
Supraglottic	92 (71–98)	87 (65–96)	65 (45–79)	32 (17–49)
T1 (29/31)				
T2 (39/39)	75 (57–86)	71 (53–83)	61 (43–74)	45 (29–60)
T1 + T2 (68/70)	82 (69–89)	78 (64–86)	62 (50–73)	39 (28–50)
Glottic	96 (93–98)	94 (90–97)	82 (76-86)	60 (54–66)
T1 (238/243)				
T2 (80/80)	89 (80–95)	82 (70–89)	79 (68–86)	54 (42–64)
T1 + T2 (318/323)	95 (91–97)	91 (87–94)	81 (76–85)	59 (53–64)

* Differences in numbers of patients between OS and DSS due to unknown causes of death.

**Table 3 cancers-13-00568-t003:** Resection status after laser surgery over T category.

T Category	R0	Rx	R1	R2	R Missing
T1 (*n* = 31)	14 (45.2%)	10 (32.3%)	5 (16.1%)	0 (0.0%)	2 (6.5%)
T2 (*n* = 39)	20 (51.3%)	8 (20.5%)	9 (23.1%)	0 (0.0%)	2 (5.1%)
T3 (*n* = 16)	4 (20.5%)	8 (50.0%)	2 (12.5%)	1 (6.3%)	1 (6.3%)
T4 (*n* = 10)	2 (20.0%)	2 (20.0%)	3 (30.0%)	2 (20.0%)	1 (10.0%)

**Table 4 cancers-13-00568-t004:** Adjuvant treatment after TLM over T category. No adj. = No adjuvant therapy.

T Category	No adj. (*n* = 47)	aRT (*n* = 43)	aCRT (*n* = 4)	aCT (*n* = 2)
T1 (*n* = 31)	23 (74.2%)	7 (22.6%)	0 (0%)	1 (3.2)
T2 (*n* = 39)	14 (35.9%)	24 (61.5%)	1 (2.6%)	0 (.0)
T3 (*n* = 16)	8 (50.0%)	6 (37.5%)	1 (6.3%)	1 (6.3)
T4 (*n* = 10)	2 (20.0%)	6 (60.0%)	2 (20.0%)	0 (0.0)

**Table 5 cancers-13-00568-t005:** Procedural characteristics by stage.

Procedure	Category	Stage I	Stage II	Stage III	Stage IV	All
ND	No	14 (63.6)	15 (39.5)	15 (28.8)	17 (20.7)	61 (31.4)
	Yes	8 (36.4)	23 (60.5)	37 (71.2)	65 (79.3)	133 (68.6)
Resection	R0	10 (45.5)	21 (55.3)	24 (46.2)	28 (34.1)	83 (42.8)
	Rx/R1/R2	11 (50.0)	9 (23.7)	15 (28.8)	37 (45.1)	72 (37.1)
	Missing	1 (4.5)	8 (21.1)	13 (25.0)	17 (20.7)	39 (20.1)
Adj. treatment	No	21 (95.5)	29 (76.3)	28 (53.8)	34 (41.5)	112 (57.7)
	Yes	1 (4.5)	9 (23.7)	24 (46.2)	48 (58.5)	82 (42.3)

**Table 6 cancers-13-00568-t006:** Success and failure, respectively, after aRT in T2 supraglottic cancer patients with a clear indication for adjuvant treatment. Each patient defined by his resection (R) and lymph node (N) status in bold letters, followed by OS [y=years, m=months], and cause of death; LT-S = long-term survivor, i.e., living at the end of follow-up time; tu = death due to locoregional recurrent tumor; NSCLC = Non-small cell lung cancer; Rect-Ca = Carcinoma of the rectum; Colon-Ca = Colon cancer; Pneumo = Pneumonia.

Indication	Success	Intermediate/Uncertain	Failure
Clear indication (R1, any N R2_,_ any N any R, N2 any R, N3)	**R0N2_b_**:13y10m, NSCLC **R0N2_b_**: 5y9m, Rect-Ca **R1N0** (aCRT!): LT-S **R1N0**: LT-S **R1N2_c_**: LT-S **R1N1**: 5y1m, Colon-Ca **RxN2_b_**: 8y7m, Pneumo	**(R0N2_b_)**: 4y3m M1	**R0N2_c_**: 3y2m, tu **R0N2**: 2y11m, tu **R0N2_c_**: 1y5m, tu **R1N0**: 0y9m tu **R1N2_b_**: 2y10m, tu **R1N2_c_**: 3y3m, tu **R1N1**: 1y5m tu **RxN2**: 1y3m, tu **RxN3**: 1y5m, tu

**Table 7 cancers-13-00568-t007:** 5-year OS of early-stage supraglottic cancer patients after TLM in different studies.

Study	Number of Patients	5-y OS [%]
Ambrosch, 1998 [17]	48	76.0 (pT_1_ + T_2_)
Iro, 1998 [18]	69	75.4 (stage I + II)
Steiner, Ambrosch, 1996 [22]	43	72.8 (stage I + II)
Ambrosch, 2018 [20]	67	69.0 (stage I + II)
Dyckhoff, present study	70	62.0 (pT_1_ + pT_2_)
Iro, 2011 [14]	137	59.4 (pT_1_ + pT_2_)

**Table 8 cancers-13-00568-t008:** Differential matrix for the definition of success of adjuvant treatment.

**Failure**	Early failure: tumor-related death ≤ 2 years	
	Intermediate failure: tumor related death 2 years < ... ≤ 5 years	
**Intermediate/** **uncertain**	Late failure/success not sustained: tumor related death > 5 years	
	Uncertain success	Death due to intercurrent disease, e.g., myocardial infarction, stroke ≤ 5 years
		Tumor-related death which cannot be prevented by locoregional treatment, i.e., distant metastases ≤ 5 years
**Success**	Sustained oncologic success: no recurrence, death due to intercurrent disease > 5 years	
	Clear success: no recurrence and long-term survivor (beyond time of follow-up)	

## Data Availability

The datasets generated and analyzed during the current study are available from the corresponding author on reasonable request.

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
