# Peer review of "An Observational Cohort Study on 194 Supraglottic Cancer Patients: Implications for Laser Surgery and Adjuvant Treatment"

_cancers, 2021, doi:10.3390/cancers13030568_

Round 1

Reviewer 1 Report

Dear editor and the members of the editorial board,

The investigators of the manuscript with the ID cancers-1055558 present the results of a relatively large cohort of patients with supraglottic laryngeal cancer and address a topic which lacks adequate number of papers in the literature so far. In principle, I would like to endorse the publication of this paper. However, the manuscript should to be substantially edited, especially in terms of the interpretation of the data. The quality of writing and the use of English language can also be improved by a native English speaker who is familiar with medical writing. In summary, this paper has a great potential just because of the subject matter and the size of the cohort. Unfortunately, in its current form this potential is being sacrificed to the futile effort of justification of primary surgery which is being praised as the treatment of choice over the inferiority and futility of radiotherapy declared in a retrospective chart review harboring unavoidable inherent strong selection bias. Below you can find my comments and suggestions in detail.

Many thanks and kind regards

- The title ends with « …in early stage ». However, the paper includes all stages. This discrepancy may be misleading for potential readers. Either the title has to be amended or Stage III-IV patients should be excluded from the study.

- The authors emphasize the terms “every-day” and “real world” in the title and throughout the manuscript. Although this is a nice way of saying “A retrospective(?) chart review with a strong possibility of harboring selection bias and confounding factors”, I would strongly suggest to refrain the authors from using these terms. It seems especially contradictory, when conductance of a prospective randomize study (i.e. an “artificial world” study?) for the validation of these results are presented as the correct methodology in the last paragraph of the Discussion section.

- It should be clearly stated, whether this was a prospective cohort study or a retrospective chart review.

- The median follow-up should be provided under the Results section and indicated whether it includes patients with recurrence and/or death or not.

- From the distribution of primary treatment modalities, it becomes quite clear, that the centers in which the treatments were performed had a very strong preference for primary surgery over primary radiotherapy +/- chemotherapy. This is a clear source of bias if a comparison between treatment modalities are indeed to be made. I assume that patients with co-morbidities, lower compliance, substance abuse and lack of social support rather underwent non-surgical treatment (and maybe de-intensified?). Actually, this is the main reason why this paper should rather focus on simply describing the experience and the results without trying to compare the results of treatment modalities via inferential statistics. It would be valuable enough just to plainly publish the results due to the fact that there is few literature exclusively about subraglottic laryngeal published so far.

- Some abbreviations are not explained in the footnotes of the tables, and have to be added.

- Table 1: The general UICC stage should be added. The procedural characteristics such as neck dissection, resection status and adjuvant therapy should be provided based on stage as well (at least as supplementary material). This is crucial, because the adjuvant treatment rate of 51% for patients who underwent primary surgery is quite high. It would be acceptable if this was dominated by loco-regionally advanced stages. Otherwise, it would show that so many patients underwent unnecessary multimodal treatment in early stage.

- Kaplan-Meier plots require the numbers at risk below their X-axes.

-  Does the data contain enough information to demonstrate laryngectomy-free survival (general and for each stage T-stage)? This endpoint has paramount importance in this patient population.

- Without addressing each detail: The authors interpret the data through the Results and Discussion sections and come up (in my understanding) with the take home message, that 1) the adjuvant radiotherapy cannot compensate poor R; and 2) primary surgery is superior than primary radiotherapy.

1) There is no/less need to compensate poor R if patients can be better selected for primary surgery without the need of adjuvant treatment or for primary radio(chemo)therapy. It is not meaningful to generate iatrogenic hypoxic conditions and declare radiotherapy as a futile measure. The whole concept of “radiation efficiency” in this paper does not make much sense. Furthermore, it should be obvious that more complicated tumors with poor prognostic factors had to receive adjuvant treatment. Methodologically it is not sound to compare apples to oranges. “Centers of excellence”, experience and patient caseload with the emphasis on surgery are mentioned in the manuscript. Of course, these are important. But, does that only mean surgical/technical expertise or does it apply to the whole diagnosis and management concept? were all patients discussed on multidisciplinary tumorboards dedicated for the diagnosis and treatment of head and neck cancer? What about the radiotherapy? Were these patients also treated in academic/teaching hospitals by radiation oncologists dedicated to head & neck cancer, or were they “general radiation oncologists”? What percent of patients undergoing adjuvant radiotherapy adhered to the package time from surgery to the end of the radiotherapy? How were their case volume? There is mounting evidence showing significantly PFS and OS results in radiation oncology clinics with higher case volume, which participate to prospective clinical trials and academic centers.

2) Except for advanced T4 of poorly-functioning T3, there are no properly designed studies (with no or minimized bias) showing any superiority of radio(chemo)therapy or primary surgery +/- adjuvant therapy over another. The main aims of larynx surgery are and remain A) keep the number of treatment modalities at minimum without compromising from oncologic outcome; B) Obtain function if feasible.

- Table 5: Personally, I find the Table 5 (and partially the Table 7) unnecessary and too complicated to interpret. If the authors still want to keep it, the patient IDs shall be removed. This does not comply with the ethical publishing standards, if these are hospital IDs indeed. R? should be replaced with Rx.

- It is interesting that the authors do not mention any possible limitations of their study in the Discussion section.

Reviewer 2 Report

This retrospective study, from 20 years ago, just confirm that transoral surgery (laser resection for supraglottic tumors) is one of the standard treatments, for early stages T1-T2 and some small T3 tumors. open surgery is the other option. R0 is better than R1 or Rx for DSS. A new surgery is mandatory if first resection appears to be R1 or Rx, and if a new surgery is not feasible, or refused by patient adjuvant chemo RT is the treatment of choice.

Everything described in this paper is already considered as a standard. For adjuvant Chemo RT this has been validated in two big phase III trials, and in a pooled analysis of both.

This paper does not add any new arguments for the treatment of supraglottic laryngeal tumors.

Round 2

Reviewer 1 Report

Dear editor-in-chief,

I would like to thank the Cancers editorial board and the authors for the processing and the extensive editing of the manuscript, which clearly improved its quality. Therefore, I endorse its publication in Cancers.

Kind regards

Author Response

Reply to Reviewer 1

Dear Reviewer,

Thank you again for your very kind and instructive review. You have helped us a lot to improve our manuscript substantially.

In our last check, we had some minor corrections which are listed below. We had the American Journal Experts to check for the spelling and hope that everything is fine now.

With kind regards and many thanks!

Gerhard Dyckhoff

Line 192: protected space after „n=20“

Line 196/97: Table 1, table-caption:  corrected to format of MDPI_4.3_table_caption

Line 201-203: Table 1, table_footer: corrected to format of MDPI_4.3_table_footer

Line 229/230 Table 2: „Supraglottic“ as heading, „T1“ next line, indented

Line 239: protected space between „95%“ and „CI“

Line 243: Table 3: a point is added at the end of the footer.

Line 261/262: Table 4: 23 (74.2%) (a period instead of the decimal comma!)

Line 369 Table 7: a point is added at the end of the footer.

Line 575: added „/or“ in the phrase: „with positive resection margins (R+) and/or extracapsular spread (ECS+) in lymph node metastases“

Line 663/449 Table 8: throughout the table „years“ instead of „y“

Reviewer 2 Report

Dear authors

After second review, we consider that authors gave some informations and clarifications, for the readers, to understand their point of view, and also to understand the message ""supraglottic tumor is specific"

I think that, nevertheless, this retrospective study is not sufficient to conclude that adjuvant chemo RT should be recommended outside the recognized recommandations : R1 margins and or ECS +, other situations need to be validated in a prospective study, dedicated to supraglottic tumors. The conclusion for this part needs to be reviewed.

This retrospective study, does provide informations on long term survival, 10 y OS reported. This is a very important point.

thank you for the clarifications you gave.

Author Response

Reply to Reviewer 2

Dear Reviewer,

Thank you for your kind comments and valuable suggestion! Indeed, we have to be cautious to recommend adjuvant chemo RT outside the recognized recommendations. Our study is retrospective and the number of our patients is very limited. Thus, further prospective studies on the special entity of supraglottic cancers is warranted. However, in our study, less than half of the failure patients had a positive resection status (R1) and in none of the patients positive extracapsular spread was reported. Thus, in the entity of supraglottic cancers, patients could experience a survival benefit even if these standard high-risk criteria are not given. We have implemented your considerations as follows:

“According to Cooper and Bernier, for HNSCC in general, there is a survival benefit for high risk patients, leading to the recognized recommendation of aCRT in patients having R+ or ECS+. In our data, less than half of the failure patients had a positive resection status (R1) and in none of the patients positive extracapsular spread was reported. Thus, in the entity of supraglottic cancers, patients could experience a survival benefit even if these standard high-risk criteria are not given. Therefore, provided that the patient is fit enough, we suggest chemoradiotherapy for supraglottic cancers in the primary setting at the early stage and in the adjuvant setting for cases at high risk both in the sense of R+ or ECS+ and simply for belonging to this tumor entity with poor prognosis. However, the presumed survival benefit has to be verified in prospective clinical studies.”

We hope that our explanatory statement and this more cautious way of recommendation find your approval.

With kind regards

Gerhard Dyckhoff
